# miR-373-3p Regulates Invasion and Migration Abilities of Trophoblast Cells via Targeted CD44 and Radixin

**DOI:** 10.3390/ijms22126260

**Published:** 2021-06-10

**Authors:** Hyun-Jung Lee, Seung Mook Lim, Hee Yeon Jang, Young Ran Kim, Joon-Seok Hong, Gi Jin Kim

**Affiliations:** 1Department of Biomedical Science, CHA University, Seongnam 13488, Korea; endeavor821@gmail.com (H.-J.L.); lsmook17@gmail.com (S.M.L.); dsyb012@naver.com (H.Y.J.); 2CHA Bundang Medical Center, Department of Obstetrics and Gynecology, Seongnam 13496, Korea; happyirang@chamc.co.kr; 3Department of Obstetrics and Gynecology, Daerim St. Mary’s Hospital, Seoul 07442, Korea; hjsobgy@gmail.com; 4Research Institute of Placental Science, CHA University, Seongnam 13488, Korea

**Keywords:** preterm labor, miR-373-3p, CD44, radixin, invasion, trophoblast cells

## Abstract

Preterm labor (PTL) is one of the obstetric complications, and is known to be associated with abnormal maternal inflammatory response and intrauterine inflammation and/or infection. However, the expression of specific miRNAs associated with PTL is not clear. In this study, we performed combination analysis of miRNA array and gene array, and then selected one miRNA (miR-373-3p) and its putative target genes (CD44 and RDX) that exhibited large expression differences in term and PTL placentas with or without inflammation. Using qRT-PCR and luciferase assays, we confirmed that miR-373-3p directly targeted CD44 and RDX. Overexpression of miR-373-3p reduced the migration and invasion of trophoblast cells, while inhibition of miR-373-3p restored the migration and invasion abilities of trophoblast cells. Finally, we validated the expression of miR-373-3p and its target genes in clinical patients’ blood. miR-373-3p was increased in PTL patients’ blood, and was the most expressed in PTL patients’ blood with inflammation. In addition, by targeting the miR-373-3p, CD44 and RDX was decreased in PTL patients’ blood, and their expression were the lowest in PTL patients’ blood with inflammation. Taken together, these findings suggest that miR-373-3p and its target genes can be potential biomarkers for diagnosis of PTL.

## 1. Introduction

The placenta, a maternal–fetal interface organ, is important for fetal development and protection and maintenance of pregnancy, because various pregnancy-related hormones, growth factors, nutrients and waste products are transported through the placenta [1]. Preterm labor (PTL) is one of the most common causes of birth before 37 weeks of gestation. Preterm birth is closely associated with infant mortality and morbidity. Obstetric complications such as PTL [2] are known to be associated with aberrant maternal inflammatory response and intrauterine inflammation and/or infection. Although intrauterine microbial invasion is the only clearly defined cause of PTL, the specific mechanisms that cause inflammatory reactions to maternal part or placenta are not fully understood. Overcoming inflammation during pregnancy can be one of the solutions to prevent PTL. Research focusing on bioactive factors secreted from placenta to maternal serum have been performed. However, most of the placental biomarkers have not shown applicable diagnostic capability [3]. Trophoblasts are major cell types of the placenta and they have an invasive capacity that acts an important role in early implantation and placental development [4]. Accordingly, regulation of invasion ability of trophoblast cells may be a solution to increase rates of implantation and maintenance of pregnancy.

MicroRNAs (miRNAs) are 18–24 nucleotides single-stranded noncoding RNA that can downregulate gene expression and protein coding through pairing with 3′ untranslated region (3′ UTR) of target messenger RNAs (mRNAs) [5]. The regulatory role of miRNAs is associated with diverse cellular processes, such as cell proliferation, invasion, apoptosis, immune-response, organ development and tumorigenesis [6,7]. A previous study [8] showed that target genes of the same miRNA can be classified into several groups, and different groups of miRNA target genes are co-expressed under different conditions. These groups of miRNA target genes are also often co-regulated. For this reason, miRNA profiling alone is controversial in predicting intracellular events, and is no longer sufficient. Therefore, although miRNA-target gene interactions have been studied, it is essential to validate their functionality in the biological model of interest because of their cell type specificity.

miR-373-3p is mainly known to regulate multiple cellular processes in tumors, and miR-373-3p enhances cell migration and invasion of tumors [9,10]. In addition, it is reported that breast cancer cell progression is increased by miR-373-3p through the post-transcriptional regulation of CD44 [11]. CD44 is a ubiquitously expressed transmembrane glycoprotein receptor [12]. The cytoplasmic tail of CD44 also binds to the ERM protein [13,14] composed of Ezrin, Radixin (RDX), and Moesin. RDX, another target protein of miR-373-3p, is a cytoskeletal protein that acts to link actin to the plasma membrane in actin-rich cell surface structure. However, the potential role of CD44 and RDX in trophoblast cells has not been elucidated, and the function of miRNA-targeting two genes is also not clear. Two important functions of trophoblast cells are migration and invasion ability [15]. Therefore, it is necessary to study the potential function of miR-373-3p and the miRNA-target genes in the migration and invasion of trophoblast cells, which will help to understand mechanisms of implantation and placental development.

## 2. Results

### 2.1. miRNA-373-3p and Its Target Genes Are Selected by miRNA and Gene Array Combination

To select miRNAs and its target genes, we analyzed miRNAs and gene array from total RNA of term and preterm placenta tissues according to inflammation. Results of the array that displayed higher and lower over a 1.5-fold difference in expression were shown in a heat-map (Figure 1A,B). The Venn diagram showed differentially expressed miRNAs between inflammatory term placentas and PTL placentas compared with normal term placentas (Figure 1C, left). Among them, miR-373-3p markedly changed by more than 1.5–fold both inflammation and PTL placentas compared with normal term placentas. To identify target genes of miR-373-3p, mirBASE and microRNA.org database was used. CD44 and RDX as putative targets of miR-373-3p are identified (Figure 1C, right). Interestingly, CD44 was remarkably upregulated in term placenta with inflammation, and the expression of RDX was decreased in PTL placenta, regardless of inflammation, compared to term in non-inflammatory placenta, via gene array results. Furthermore, the expression of miR-373-3p negatively correlated with CD44 or RDX in inflammatory term placenta or PTL, respectively. 

### 2.2. miRNA-373-3p Targets Two Genes, CD44 and RDX at the Same Manner in Placenta as Well as Trophoblast Cells

To validate the expression of miR-373-3p, we conducted qRT-PCR from RNA in the placental tissues. Expression of miR-373-3p in the inflammatory term placentas was decreased compared with those of non-inflammatory placentas (*p* < 0.05, Figure 2A). To identify expression of target genes, CD44, we conducted qRT-PCR and Western blot in the term placentas with or without inflammation. As shown in Figure 2B,C, the RNA expression of CD44 was significantly increased in the inflammatory term placentas compared with the non-inflammatory placentas, but the protein level was slightly decreased, unlike RNA expression. In the case of non-inflammation, the expression of miR-373-3p in the PTL placentas was also increased compared to in term placentas (*p* < 0.05, Figure 2D). Unlike CD44, both RNA and protein expression of RDX were decreased in the PTL placentas. These results suggest that miR-373-3p targets two proteins, CD44 and RDX, and suppresses their expression in each condition (Figure 2E,F). To confirm localization of CD44 and RDX in placental tissues, we conducted immunofluorescence. As shown in Figure 2G, CD44 (green) and RDX (red) were expressed in the non-inflammatory term placental tissue. In a negative control, interestingly, the expression of RDX as well as CD44 was gradually reduced in the order of inflammatory term and PTL placentas, similar to the Western blot results. This means that expression of CD44 and RDX is more decreased in PTL placenta than term placenta with or without inflammation.

### 2.3. miR-373-3p Targets CD44 and RDX Directly and Downregulates Their Expression

To investigate whether miR-373-3p inhibits CD44 and RDX by directly targeting complementary sequences on the 3′UTR regions of mRNAs, we made constructs with 3′UTR of its target genes and used a luciferase miRNA assay. Luciferase miRNA target expression vector is designed to quantitatively evaluate miRNA activity by inserting miRNA target sites of the firefly luciferase gene (luc2). These target sites can be introduced by cloning each 3′ UTR of a gene of interest (e.g., CD44 or RDX), to study the influence of these sites on transcript stability and activity. Firefly luciferase is the primary reporter gene, and reduced firefly luciferase expression indicates the binding of introduced miRNAs to the cloned miRNA target sequence. So, each clone, including 3′UTR of CD44 and RDX, was transfected with a mimic of miR-373-3p into Hela cells. We observed that the miR-373-3p mimic moderately but significantly repressed the reporter activities by binding to each 3′UTR region of CD44 and RDX, compared with mock or nonspecific miRNA (NC) transfected cells as control groups. This means that miR-373-3p specifically reacts to 3′UTR of CD44 and RDX (*p* < 0.05, Figure 3A,B).

### 2.4. miR-373-3p Negatively Regulates the Expression of CD44, RDX, and Adhesion Molecules

To further investigate the effects of miR-373-3p overexpression on CD44 and RDX, we transfected mimic or inhibitor of miR-373-3p into HTR8/SVneo cells and observed changes in CD44 and RDX mRNA levels. Expression of miR-373-3p was significantly increased in mimic-transfected cells and lowest in inhibitor-transfected cells (*p* < 0.05, Figure 4A). As expected, mRNA expression of CD44 and RDX was decreased by miR-373-3p mimic transfection and restored to the control level by miR-373-3p inhibitor (Figure 4B). Although not as dramatic as the RNA levels, changes in CD44 protein expression by miR-373-3p mimic or inhibitor transfection were similar to those of mRNA, but little change in protein expression of RDX was observed (*p* < 0.05, Figure 4C,D). These results suggest that miR-373-3p downregulated CD44 and RDX primarily at the mRNA rather than protein levels. During the process of implantation, vascular invasion by trophoblast is regulated by adhesion molecules that interact between endovascular trophoblast and decidual endothelial cells [16]. As shown in Figure 4E, the expression of ITGβ7, p-FAK, Rock-1 as integrin-related adhesion complex, was markedly decreased in miR-373-3p overexpressed cells, and slightly recovered by miR-373-3p inhibition. However, ITGα5 and Rho A did not show significant changes at the protein level by miR-373-3p mimic or inhibitor treatment. Rac1 did not show a difference in protein level by miR-373-3p mimic, but its expression was rather decreased by miR-373-3p inhibition (Figure 4E).

### 2.5. The Migration and Invasion Ability of Trophoblast Cells Was Inhibited by Down-Regulation of CD44 and RDX by Targeting of miR-373-3p

We additionally investigated the potential role of miR-373-3p on migration and invasion of trophoblast cells, through wound healing assay and invasion assay, respectively. At 24 h after miR-373-3p mimic transfection, there was no significant effect on migration rate, compared with the control cells, but a significant decrease was observed after 48 h. On the other hand, miR-373-3p inhibitor transfection increased HTR8/SVneo cells migration in a time-dependent manner (*p* < 0.05, Figure 5A,B). In addition, the invasion ability of HTR8/SVneo cells through transwell chamber was found to be consistent with the results of wound healing assay (*p* < 0.05, Figure 5C). The images show the invaded cells (upper panel), and the graph shows the number of invading cells to the bottom of insert (lower panel). We further investigated whether miR-373-3p affects the expression of MMP-9, an enzyme that degrades extracellular matrix. Samples for zymography were harvested from the supernatant of the insert used in the invasion assay. Similar to the result of invasion assay, enzyme activity of MMP-9 decreased in miR-373-3p mimic-transfected cells and increased in miR-373-3p inhibitor-transfected cells (Figure 5D). It means that miR-373-3p is a negative regulator of trophoblast migration and invasion.

### 2.6. Expression of miR-373-3p, CD44 and RDX in Normal and PTL Patients’ Blood

As we confirmed the expression of miR-373-3p and CD44 and RDX in term and PTL placenta tissues, we conducted qRT-PCR analysis to validate their expression in clinical normal and PTL patients. miR-373-3p is more increased in PTL patients’ blood than that of normal patients, and its expression is much more increased by inflammation (Figure 6A). By targeting miR-373-3p, CD44 and RDX decrease more in PTL patients’ blood than in that of normal patients, and their expression is more decreased by inflammation (Figure 6B,C). These results coincide with results with placenta tissues (Figure 2D,E) except for CD44, and it represents the validated results with clinical patients’ blood as well as placenta tissues. Therefore, it represents the potentials of miR-373-3p as a specific biomarker which could detect PTL.

## 3. Discussion

MicroRNAs have potential as biomarkers for diagnosis of diseases [17], but related studies are not sufficient to investigate the potential roles of miRNAs in molecular regulatory mechanisms, as well as the fact that there are still controversies on their roles and functions. Since miRNAs have the ability to control the expression of various genes at the same time [18], their expression patterns of miRNAs have limitations in explaining the onset mechanism and regulatory mechanism of the disease. For this reason, it is necessary to understand the function of miRNA in order to identify the correlation among various genes targeted by miRNA.

Preterm labor occurs due to several obstacles such as inflammation as well as blood supply into the placenta, as trophoblast cells are not penetrated into maternal portion [19]. Therefore, we need to explain the cause of PTL in functional aspects of trophoblast cells by miRNAs expression. Interestingly, we found that miR-373-3p simultaneously target two genes, CD44 and RDX, through the combination between miRNA array and gene array in placental tissues and trophoblast cells. Recent studies suggest that expression of miR-373-3p is increased in various cancers including lung cancer, breast cancer and gastric cancer. Although the role of miR-373-3p is well-known in cancer cells [9,11,20,21], it is not elucidated that the regulatory mechanism of miR-373-3p and the effect of its target genes on trophoblast cells are related to placental development. In addition, miR-373-3p increased in the PTL placenta compared to the term placenta, regardless of inflammation, and then downregulated CD44 and RDX as its target genes. Overexpression of miR-373-3p in vitro experiments using trophoblast cells reduced expression of adhesion molecules as well as CD44 and RDX. Sequentially, it inhibits the invasion and migration of trophoblast cells through downregulation of MMP-9 [22].

The reason why the mechanism acts in this way could be explained, because the crystallographic binding between FERM of RDX and cytoplasmic long tail of CD44 has been reported in a previous study [14], although precise details concerning the interaction are unknown. Therefore, one miR-373-3p simultaneously targets each CD44 and RDX, then the complex which is a result of the interaction between CD44 and RDX is regulated by targeting of miR-373-3p. Then, the regulated CD44-RDX complex by miR-373-3p could regulate the signaling of downstream related to adhesion, migration and invasion in trophoblast cells.

Generally, miR-373-3p acts as a potent proto-oncogenic factor in tongue squamous cell carcinoma (TSCC), and upregulation of miR-373-3p leads to the promotion of TSCC invasion via activation of Wnt signaling by the downregulation of Dickkopf-1 (DKK1) [20]. This means that, although the precursor miRNA is the same, the targeting properties are different by alternative mature products [23,24] and function of miRNA could also be different according to circumstance such as cell type, species, and targeting genes [24]. These similar results suggest that miRNA-373-3p may be involved in the migration and invasiveness of miRNA-373-3p by controlling the infiltration ability of trophoblast cells, as well as controlling the infiltration ability of the miRNA-373-3p in cancer cells. In addition, the mRNA-protein expression correlation was relatively inconsistent. A previous study reported by Bartel et al. [25] showed that changes in mRNA levels closely reflect the impact of miRNAs on gene expression, and indicate that instability of target mRNAs is the main reason for reduced protein production [26]. In support of this “mRNA-destabilization” theory, our results also showed that overexpression of miR-373-3p leads to a markedly inhibitory effect of CD44 and RDX, whereas miR-373-3p inhibition is completely restored at the mRNA level of CD44 and RDX, but is incomplete at the protein levels.

In addition, the correlation between miRNA and target genes is slightly worked, but we could observe that the invasion and migration abilities of trophoblast cells are restored by the inhibition of miR-373-3p. Finally, we confirmed that miR-373-3 is highly expressed in clinical PTL patients’ blood, similar to the results from PTL placenta tissues. As placenta tissues are organs which are obtained after delivery, the results from the tissues have limitations as a specific marker for diagnosis of diseases. Therefore, the results from clinical PTL patients’ blood before delivery could be a basis for predicting the possibility of the onset of PTL. Thus, we demonstrated the analysis of miRNA profiling and cDNA profiling, which select matched factors and regulate the infiltration ability of trophoblast cells by selectively regulating CD44 and RDX expression in selected miRNA-373-3p, and their regulation can affect placental development as well as obstetrical diseases, including preterm labor.

In conclusion, we demonstrated that miR-373-3p simultaneously targets two different genes, CD44 and RDX, in placenta tissue and trophoblast cells. Therefore, the expression of CD44 and RDX is decreased, which leads to the inhibition of migration and invasion of trophoblast cells by downregulating MMP-9 including adhesion molecules. Therefore, these data suggest that miR-373-3p has potential as a biomarker for the diagnosis of PTL, because miR-373-3p affects the function of trophoblast as an important cell during placental development, and is detectable in clinical patients’ blood. In addition, these findings help us to understand underlying mechanisms of miRNA function and regulatory mechanisms of trophoblast cells through miRNAs in placental development.

## 4. Materials and Methods

### 4.1. Placenta Collection

The collection of placenta tissues and their use for research purpose were approved by the Institutional Review Board of Seoul National University College of Medicine (H-1105-045-361), Seoul, Korea. All patients provided written informed consents prior to sample collection. Term placentas were collected from women who delivered at term (37 ≥ gestational weeks) without any pathological problems. Preterm labor (PTL) placentas were collected from women who delivered at preterm (37 < gestational weeks). The placental tissues were sub-divided into four groups according to the pathological status. The placentas from patients diagnosed with chorionitis and choriodeciduitis were referred to as inflammation (+) groups such as term (+) and PTL (+) (Table 1). Right after delivery, the placenta tissues were sampled according to the purpose of analysis within 1 h. Placenta tissues were obtained from the central area of the placentas and snap frozen in liquid nitrogen for RNA and protein isolation. In addition, placenta tissues were frozen in OCT compound for immunofluorescence. All samples were stored at −80 °C until analysis.

### 4.2. miRNA and mRNA Microarray and Computational Analysis

Total RNA was extracted from the human placenta tissues using the TRI reagent (MRC, Cincinnati, OH, USA) according to the manufacturer’s instructions. The quality of RNA was assessed by Agilent Bioanalyzer 2100 analysis. For miRNA expression profiling, total RNA was poly(A)-tailed and labeled with FlashTag biotin HSR labeling Kit (Affymetrix, Santa Clara, CA, USA). The labeled RNA was hybridized with the Affymetrix miRNA 4.0 array for (Affymetrix) 16 to 18 h at 48 °C. Biotinylated cRNA was prepared for mRNA expression profiling. Following fragmentation, 12 ug of aRNA were hybridized for 16 h at 45 °C on GeneChip^®^ Affymetrix Primeview array (Affymetrix). After being hybridized and washed, the arrays were scanned using the Affymetrix GeneChip Scanner 3000 7G. The data were analyzed with Robust Multichip Analysis (RMA) using Affymetrix default analysis settings and global scaling as normalization method. The trimmed mean target intensity of each array was arbitrarily set to 100. The normalized, and log-transformed intensity values were then analyzed using GeneSpring GX13.0 (Agilent technologies, Santa Clara, CA, USA). Fold change filters included the requirement that the genes be present in at least 150% of controls for upregulated genes and lower than 66% of controls for downregulated genes. Hierarchical clustering data were clustered into groups that behaved similarly across experiments using GeneSpring GX 13.0 (Agilent technologies). The clustering algorithm was Euclidean distance, average linkage.

### 4.3. Cell Culture

HTR-8/SVneo cell line (human first trimester cytotrophoblasts, CTBs) was provided by Dr. Charles H. Graham (Queen’s University, Kingston, ON, Canada). Hela cells were purchased from American Type Culture Collection (ATCC, OT, Canada). HTR-8/SVneo cells were maintained in RPMI medium 1640 (Hyclone, Logan, UT, USA) supplemented with 5% fetal bovine serum (FBS, GIBCO, Carlsbad, CA, USA) and 100 U/mL penicillin and streptomycin (GIBCO). Hela cells were maintained in DMEM-low glucose medium (GIBCO) supplemented with 10% FBS and 100 U/mL penicillin and streptomycin (GIBCO). At the end of the incubation, the cell pellets were collected and maintained at −80 °C until they were needed for analysis.

### 4.4. RNA Extraction and Quantitative Real-Time PCR

Total RNA of placenta tissue samples and HTR8/SVneo cells and PTL patients’ blood were extracted using Trizol reagent (Invitrogen, Carlsbad, CA, USA) and RNeasy mini prep kit (Qiagen, Valencia, CA, USA) and miRNeasy Serum/Plasma Kit respectively. To confirm the expression of target genes, the 300 ng of RNA was synthesized using Superscript III RNase H reverse transcriptase (Invitrogen) according to the manufacturer’s protocols. For expression of miR-373-3p, the 300 ng of RNA was synthesized using Mir-X miRNA First-Strand Synthesis Kit (Clontech, Mountain View, CA, USA) according to the manufacturer’s protocols. The mRNA levels of CD44 and RDX were determined by quantitative real-time PCR using SYBR green master (Roche, Basel, Switzerland) according to the manufacturers’ protocols. Specific primers for CD44 and RDX sequences are presented in Table 2. Target gene expression was normalized with expression of GAPDH as an internal control. The specific primer for GAPDH sequences is: GAPDH-F: 5′CGA GAT CCC TCC AAA ATC AA-3′, GAPDH-R: 5′-TGT GGT CAT GAG TCC TTC CA-3′. Target sequences were amplified by using the following thermal conditions in an Exicycler 96 real-time PCR machine (Bioneer, Daejeon, Korea): 10 min at 95 °C, and 40 cycles of 10 s at 95 °C and 30 s at 60 °C (for CD44, RDX, and GAPDH), 10 min at 95 °C, and 40 cycles of 10 s at 95 °C and 2 at 60 °C and 1 min at 95 °C, 30 s at 55 °C, and 30 s at 95 °C (For miR-373-3p). All reactions were performed in duplicate.

### 4.5. Transfection of Mimic and Inhibitor of miR-373-3p

For transfection, 4 × 105 of HTR-8/SVneo trophoblast cells were seeded onto 100 mm culture dishes. According to the manufacturer’s instructions, when cells reached 70% confluence, they were transfected with 50 nM of miR-373-3p mimic and inhibitor using Lipofectamine 2000 (Invitrogen). Transfected cells were maintained in serum-reduced media, OPTI-MEM (GIBCO) for 24 h and dissolved in lysis buffer (Sigma, St. Louis, MO, USA). Treatment of only Lipofectamine in HTR8/SVneo cells was used as a negative control.

### 4.6. Plasmid Constructs and Luciferase miRNA Assay

The 3′UTR sequences were confirmed by the National Center for Biotechnology Information (NCBI) site. The fragments of CD44 and RDX 3′UTR nucleotides were PCR-amplified and then subcloned into the XbaI and Xho1 site in pmirGLO-Control vector (Promega, Madison, WI, USA). All constructs were sequence-verified (Bioneer Corporation). Hela cells (5 × 10^4^) were seeded in triplicate onto 24-well plates, and cultured in 80% confluent. Cells were transfected with pmirGLO Dual-Luciferase miRNA Target Expression plasmid (pmirGLO-CD44-3′UTR and pmirGLO-RDX-3′ UTR) with a negative control for miRNA in each combination, and the mimic of miR-373-3p using Lipofectamine 2000 (Invitrogen). Luciferase signals were measured 24 h after transfection using Dual Luciferase Reporter assay kit (Promega) according to the manufacturer’s protocol.

### 4.7. Western Blot

Term placental and PTL placental tissues were grinded and lyophilized in liquid nitrogen by mortar and pestle. Grinded placental tissues were lysed in RIPA buffer (SIGMA) with complete protease inhibitor cocktail (Roche) and phosphatase inhibitor cocktail Ⅱ (A.G Scientific Inc., San Diego, CA, USA). HTR8/SVneo cells were also lysed in the RIPA buffer with protease and phosphatase inhibitors. The protein concentration was measured using the BCA protein assay kit (ThermoFisher, Waltham, MA, USA). Equal amounts of protein were separated using 8% sodium dodecyl sulfate-polyacrylamide gels electrophoresis (SDS-PAGE) and transferred onto polyvinylidene difluoride membranes (PVDF; Bio-Rad Laboratories, Hercules, CA, USA) using a trans-blot system (Bio-Rad Laboratories). After blocking with 5% bovine serum albumin (BSA), they were incubated overnight at 4 °C with the following primary antibodies specific for CD44 (1:1000, Novus, Centennial, CO, USA), RDX (1:1000, Abcam, Cambridge, MA, USA), ITGα5 (1:1000, BD, Franklin Lakes, NJ, USA), ITGβ7 (1:1000, R&D systems, Minneapolis, MN, USA), p-FAK (1:1000 Cell signaling, Danvers, MA, USA), Rock-1 (1:1000, Cell signaling, Danvers, MA, USA), Rac1 (1:1000, Novus), RhoA (1:1000, Abcam), and GAPDH (1:5000, Abfrontier, Seoul, Korea). After washing, the blots were incubated for 1 h with horseradish peroxidase-conjugated secondary antibodies (1:20,000, Goat-anti-rabbit-IgG HRP and Goat-anti-mouse-IgG-HRP, Bio-Rad). The proteins were visualized with an ECL detection system (Bio-Rad). The intensities of the detected bands were quantified using the Image J program (NIH, Bethesda, MD, USA).

### 4.8. Immunofluorescence

To confirm the localization of CD44 and RDX in placental tissues, placental tissues were embedded in OCT compound and then sectioned on 6 μm of thickness. Placental tissues were fixed in 100% methanol for 15 min. After washing with cold-PBS, they were blocked in blocking solution (Dako, Santa Clara, CA, USA) for 30 min at room temperature. They then were incubated with rabbit anti-radixin (1:100 dilution in antibody diluent) in a humidified chamber overnight at 4 °C. Next day, the slides were incubated with Alexa 488-conjugated secondary antibody (1:200) at room temperature in the dark for 30 min. After washing, the slides were incubated with mouse anti-CD44 (1:100 dilution in antibody diluent) at room temperature in the dark for 2 h, and then incubated with Alexa 594-conjugated secondary antibody (1:200) at room temperature in the dark for 30 min. 1 μg/mL of DAPI was used as a counterstain. A negative control was used; a rabbit IgG isotype control (1:200) for RDX and a mouse IgG isotype control (1:200) for CD44, instead of each specific primary antibody (Data not shown). Images were analyzed using an EOVS fl microscope (AMG) at 200× magnification.

### 4.9. Wound-Healing Migration Assay

HTR8/SVneo cells were grown to confluency in 6-well plates and then with low-serum (0.5% FBS) media and treatment of 10 μM cyclohexamide for 6 h. Subsequently, the cell monolayer was wounded with a sterile plastic pipette tip, and then washed cells with PBS to remove detached cells and debris. The cells were transfected with mimic or inhibitor of miR-373-3p using Lipofectamine 2000 (Invitrogen) for 24 and 48 h. The photographs were captured using an optical microscope at the same time point.

### 4.10. Invasion Assay

Invasion assay was conducted using a transwell chamber with 8 μm filter insert (BD), precoated with matrigel. 5 × 10^4^ cells were seeded on the inserts in OPTI-MEM media (GIBCO). At the same time, the mimic or inhibitor of miR-373-3p was treated onto the inserts. After incubation for 24 h, the supernatant of inserts was harvested for gelatin zymography. The cells inside the inserts were swabbed by cotton buds and washed in PBS. The cells in the bottom of the inserts were fixed in 100% methanol for 10 min. After drying, they were stained with Mayer’s hematoxylin (Dako) for 8 min. The number of stained cells was randomly counted in three non-overlapping fields on the membrane.

### 4.11. Gelatin Zymography

To analyze MMP-9 activity, the supernatant harvested from upper chamber used for invasion assay was separated in 8% SDS-PAGE gel containing 0.1% gelatin (SIGMA). After running, the gel was washed in 1× renaturation buffer (Bio-Rad) for 30 min and incubated in 1× developing buffer (Bio-Rad) overnight at 37 °C. The next day, the gel was stained with Comassie Brilliant Blue R-250 in fixing solution containing 10% acetic acid and 40% methanol for 3 h. The gel was then washed in de-staining solution containing 10% acetic acid and 40% methanol until white bands were shown. MMP activity was determined by intensity of band via Image J program.

### 4.12. Statistical Analysis

The results are presented as means ± SD. Statistical significance was measured by multiple comparisons using Student’s *t*-test with a significance level of *p* < 0.05.

## Figures and Tables

**Figure 1 ijms-22-06260-f001:**
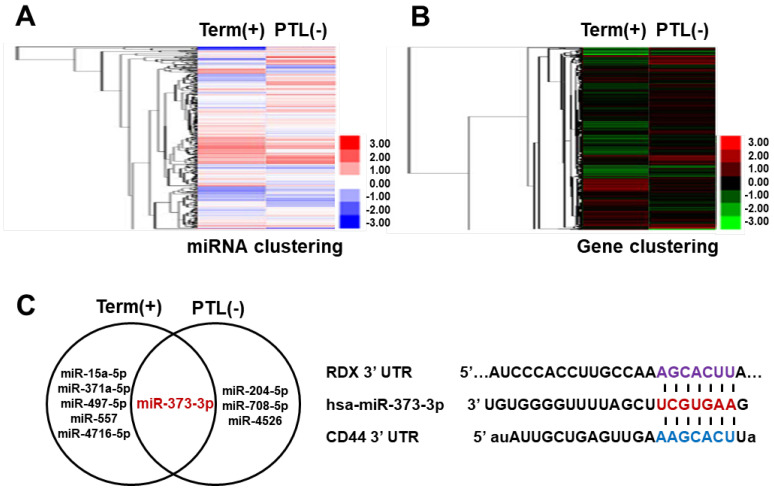
Sorting of miRNA and targeting gene via miRNA and gene array. (**A**) Clustering of *miRNA* by heat-map. Red and blue indicate high expression and low expression of *miRNA*, respectively. (**B**) Clustering of genes by heat-map. Red and green indicate high and low gene expression, respectively. (**C**) Venn diagram is shown to differentially express miRNA between inflammatory term placentas and non-inflammatory *PTL* placenta by normalizing with non-inflammatory placentas (**left**). The sequences of the *miR-373-p3* binding sites within *CD44* and *RDX 3*′*UTRs* (**right**).

**Figure 2 ijms-22-06260-f002:**
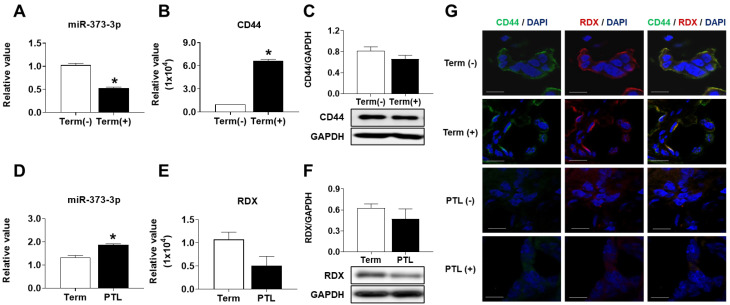
Validation of miRNA array and gene array in RNA and protein levels. Expression of *miR-373-3p* (**A**) and *CD44* (**B**) depending on inflammation in term placenta by *qRT-PCR*, and quantification of *CD44* protein expression (**C**). Expression of *miR-373-3p* (**D**) and *RDX* (**E**) in term and *PTL* placentas without inflammation by *qRT-PCR*, and quantification of *RDX* protein expression (**F**). (**G**) Co-localization of *CD44* (green) and *RDX* (red) and in placenta tissue by immunofluorescence. A negative control was used; a rabbit IgG isotype control for *RDX* and a mouse IgG isotype control for *CD44*. DAPI was used for nuclear staining as a counterstain. (Magnification, 200×). * *p* value < 0.05 compared towith Term (−).

**Figure 3 ijms-22-06260-f003:**
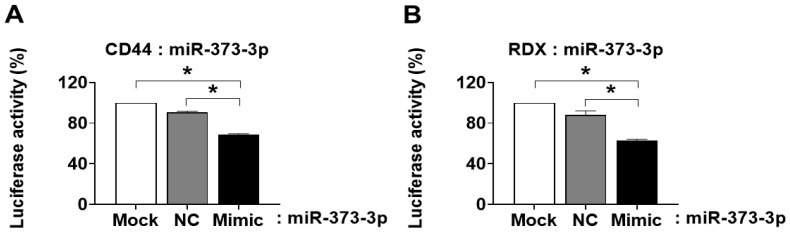
Confirmation of miR-373-3p targeting to CD44 and RDX. miR-373-3p was co-transfected with engineered luciferase reporter construct including the *CD44 3*′*UTR* (**A**) or RDX 3′UTR (**B**), harboring the target sequence of *miR-373-3p*. As a negative control, the luciferase reporter construct was engineered with a negative scrambled *miRNA* (*NC*). * *p* value < 0.05 compared with Mock or NC.

**Figure 4 ijms-22-06260-f004:**
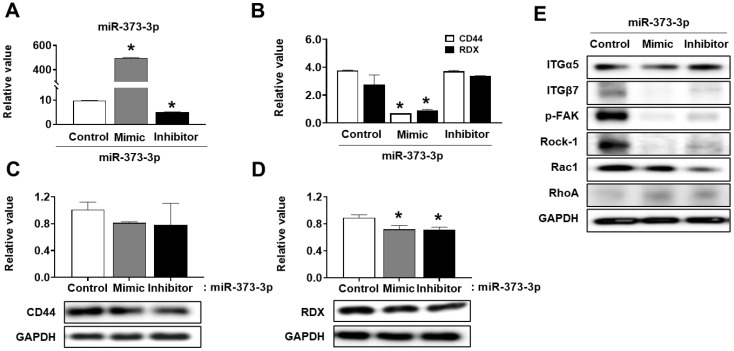
Downregulation of CD44, RDX, and adhesion molecules by miR-373-3p. Expression of miR-373-3p. (**A**) *CD44*, and *RDX* (**B**) after transfection of mimic or inhibitor of *miR-373-3* in *HTR8/SVneo* cells by *qRT-PCR*. Expression of *CD44* (**C**), *RDX* (**D**), adhesion molecules (**E**) after transfection of mimic or inhibitor of *miR-373-3* in *HTR8/SVneo* cells by Western blot. Intensity graph of band was measured by image J program through duplicate experiments. * *p* value < 0.05 compared with control.

**Figure 5 ijms-22-06260-f005:**
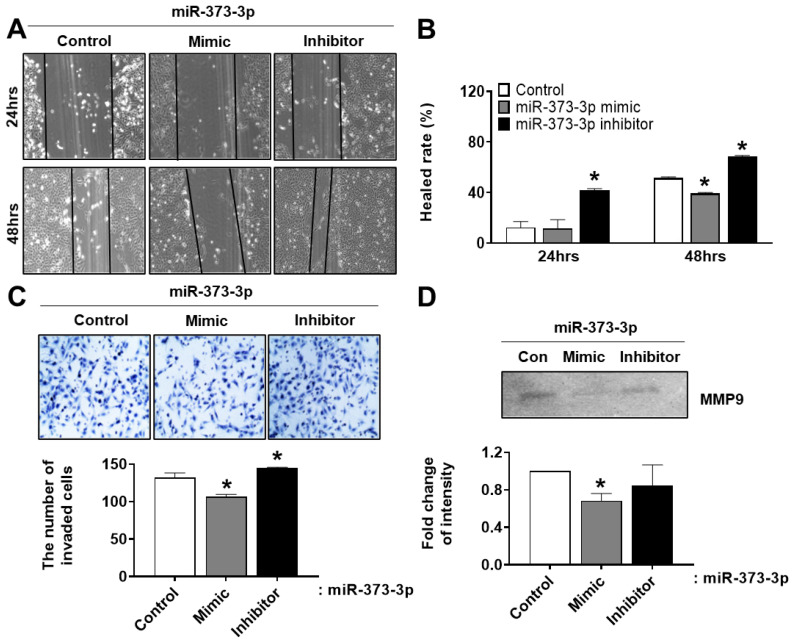
miRNA-373-3p reduces migration and invasion ability of HTR8/SVneo cells. (**A**,**B**) Wound healing assay. The images were taken using an inverted microscope and then healing area was calculated using the image J program. (**C**) Matrigel invasion assay. The total number of invading cells was calculated from the image J program. (**D**) The enzyme activity of MMP-9 was determined by zymography. Intensity graph of zymography was measured by image J program through duplicate experiments. * *p* value < 0.05 compared with control.

**Figure 6 ijms-22-06260-f006:**
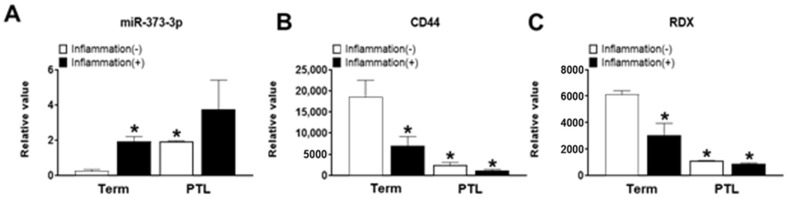
Expression of *miR-373-3p*, *CD44* and *RDX* in normal and PTL patients’ blood. Expression of miR-373-3p (**A**) *CD44* (**B**) and *RDX* (**C**) in normal and *PTL* patients’ blood depending on inflammation by qRT-PCR. * *p* value < 0.05 compared with Term without inflammation.

**Table 1 ijms-22-06260-t001:** The characteristics of maternal patients.

Variable	Term (−) (*n* = 6)	Term (+) (*n* = 2)	PTL (−) (*n* = 3)	PTL (+) (*n* = 2)
Gestational age (wk)	38.3 (37.4–38.6)	38.0 (37.1–38.6)	31.8 (26.1–35.0)	30.4 (26.0–34.6)
Birthweight (g)	2663 (2120–3170)	2180 (1900–2460)	1416 (610–2090)	1090 (540–1640)
Cesarean Delivery	5/6	2/2	1/3	0/2
Apgar score				
1 min	6 (2–9)	6.5 (6–7)	5.7 (1–9)	4.5 (3–6)
5 min	8 (5–9)	7.5 (7–8)	6.7 (2–10)	7.5 (8–7)
Pathologic result				
Chorionitis	0	0	0	1/2
Choriodeciduitis	0	2/2	0	1/2

**Table 2 ijms-22-06260-t002:** qRT-PCR primers used in this study.

Name	Sequence (5′–3′)
RDX	Forward: GAA AAT GCC GAA ACC AAT CAA
	Reverse: GTA TTG GGC TGA ATG GCA AAT T
CD44	Forward: CAA TAG CAC CTT GCC CAC AAT
	Reverse: AAT CAC CAC GTG CCC TTC TAT GG
GAPDH	Forward: CGA GAT CCC TCC AAA ATC AA
	Reverse: TGT GGT CAT GAG TCC TTC CA

## Data Availability

The data presented in this study are available on request from the corresponding author.

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
