# Peer review of "miR-373-3p Regulates Invasion and Migration Abilities of Trophoblast Cells via Targeted CD44 and Radixin"

_ijms, 2021, doi:10.3390/ijms22126260_

Round 1
Reviewer 1 Report
The manuscript uses gene profiling and mi-RNA analysis to identify and determine two targets for miR-373-3p. The manuscript needs some editing to make it easier to read. The following comments also need to be addressed:
- Why was term no inflammation chosen as the control group?
- CD44 and RDX levels are assayed in two different conditions - CD44 was never assayed in PTL. So the conclusions made about about both being targets for miR-373-3p are not clear. Both targets need to be assessed in all the pathologies/the authors need to make the reasoning for it very clear.
- In figure 2.A-C, decreased levels of miR-373-3p correlated to increased CD44 mRNA and vice versa for RDX in 2D-E. How do the authors then explain reduction in both CD44 & RDX promoters in luciferase biding assay?
- Figure 4E, densitometric analysis for the adhesion molecules is missing.
- Based on FIgure 2A, decrease in miR-373-3p should cause an increase in CD44, how do the authors explain results in Fig 4?
Author Response
Dear Reviewer,
We greatly appreciate your careful evaluation of our 1st revised manuscript (IJMS-1212687) entitled: “miR-373-3p regulates invasion and migration abilities of trophoblast cells via targeted CD44 and radixin” We were really encouraged by the reviewers’ positive comments and constructive suggestions. I am happy to report that we have successfully addressed all issues and concerns through additional data and subsequent revision of our manuscript, as detailed in the following response page. As Reviewer’s commented, we corrected it clearly stating with each comment and changes are highlighted in red in the revised manuscript,

Reviewer 2 Report
This manuscript identifies an miRNA (miR-373-3p) and two putative target genes (CD44 and RDX) expressed at elevated levels in term pregnancy and preterm labor (PTL) placentas in the presence or absence of inflammation. Direct targeting of CD44 and RDX by miR-373-3p was verified by qRT-PCR and a luciferase assay. Studies utilizing inhibition and overexpression of miR-373-3p supported an involvement of miR-373-3p in the migration and invasiveness, respectively, of cultured trophoblast cells. The expression of miR-373-3p, CD44 and RDX in the blood of normal and PTL patients’ blood relative to the presence or absence of inflammation was determined by qRT-PCR and indicated that miR-373-3p was elevated in PTL patients’ blood and was further increased by inflammation. Elevation of miR-373-3p in blood was accompanied by reduction in CD44 and RDX as assessed by qRT-PCR. The authors conclude that miR-373-3p and its target genes may be potential biomarkers for diagnosis of PTL.
There are some interesting observations presented in this manuscript. The studies appear to be adequately designed. Significant technical English editing is necessary to better communicate the findings and their interpretation. The discussion is particularly difficult to follow.
The methods section requires additional detail in support of rigor and reproducibility:
- The interval from delivery of placentae to tissue processing should be mentioned.
- Source of cell lines should be included.
- It would be helpful if the primers could be presented in a table. Similarly, a table listing primary and conjugated secondary antibodies and catalog numbers in addition to vendor source would be helpful.
- Were the antibodies used for immunofluorescence the same as for Western blot?
- Controls for immunostaining are not discussed and should be included. It might be helpful if to mention that DAPI was as a ‘nuclear” counterstain.
- As important microscope magnification given is the objective lens magnification and it’s numerical aperture.
General comments:
- Line 36 Perhaps: Morbidity and mortality of infants are associated with preterm birth.
- Line 44 Perhaps: A large part of the placenta is derived from trophoblast.
- Lines 86 – 88. Sentence is difficult to follow
- Lines 149 151. Only ITGB7 is an adhesion molecule (integrin). pFAK and Rock-1 are associated with integrin adhesion complexes. Also, GAP43 is not a marker of gap junctions rather it is Growth Associated Protein 43. Information about this protein in the context of the investigation was not discussed.
- The entire discussion is difficult to follow.
Figures:
- The Figure 2G Immunofluorescence analysis of CD44 and RDX is an important figure, however, it will be much more useful to appreciate this data if the separate green CD44/DAPI and red RDX/DAPI and CD44/RDX/DAPI overlay could be shown for the Term (-)/PTL(-) and TERM (+)/PTL(+) data.
- Figure 2. Panel F Y-axis should read RDX/GAPDH. Also, the RDX protein band intensity appears to be substantially lower and this is not reflected in the graph.
- Figure 3. Include asterisks to identify significance.
- Figure 4. Line 151. Rac1 in Figure 4E appears to be reduced by inhibitor.
- Figure 5. Wound healing assays appear to have quite a bit of cellular material in the scraped areas. Are these cells that have migrated or residual cell debris from scraping cells?
Author Response

(The authors gave the same response as above.)

Round 2
Reviewer 1 Report
Thank you for addressing the comments. I have a few further concerns:
1) Its still not clear in the manuscript as to what placentae were used and their distribution in "+"inflammation and "-"inflammation.. The authors describe it as normal without complications. And while I agree the term no inflammation is probably the right control (as outlined in the response), the explanation for the word "normal " delivery needs to be explained. Were these samples collected from women with no complication and had a cesarean delivery? Because they delivered normally with labor, they still had inflammation. Is this what the authors are trying to indicate in lines: 272-275? Is inflammatory term placentas just term with labor and normal term placenta just term with no labor? Please use these universal names for identifying the categories.
Alternatively, a table for patient characteristics outlining GA, mode of delivery etc needs to be included.
2)Was the comparison PTL(+) and term (+) used for targetscan analysis? \
3) Please add the information about the luciferase miRNA assay in the results section and change on line 126, add in luciferase "miRNA" assay as in the generic luciferase assay one does not see florescence till there is binding.
Author Response
Dear Reviewer,
We greatly appreciate your careful evaluation of our 2nd revised manuscript (IJMS-1212687) entitled: “miR-373-3p regulates invasion and migration abilities of trophoblast cells via targeted CD44 and radixin” We were really encouraged by the reviewers’ positive comments and constructive suggestions.
As Reviewer’s commented, we corrected it clearly stating with each comment and changes are highlighted in red in the 2nd revised manuscript. Please check the attached 2nd revision letter.

Reviewer 2 Report
The authors have made a number of improvements to the manuscript although they may have overlooked two important suggestions/recommendations.
- In the previous review it was noted that: “Controls for immunostaining are not discussed and should be included.” Authors should refer to Hewitt et al., Controls for Immunohistochemistry: The Histochemical Society’s Standards of Practice for Validation of Immunohistochemical Assays. J Histochem Cytochem, 2014 Oct;62(10):693-7. Most importantly, a negative control for the specificity of the primary antibody was not mentioned. An isotype-specific immunoglobulin is considered a minimum requirement.
It was noted that there was a need to obtain some assistance with technical English editing, particularly the discussion which is difficult to follow. The authors only added a single summary sentence: “So, we demonstrated the analysis of miRNA profiling and cDNA profiling, which selects matched factors and regulates the infiltration ability of trophoblast cells by selectively regulating CD44 and RDX expression in selected miR256 NA-373-3p, and their regulation can affect placental development as well as obstetrical diseases including preterm labor.” This sentence does not improve the discussion which remains difficult to follow.
Author Response

(The authors gave the same response as above.)

Round 3
Reviewer 2 Report
The authors have acknowledged the use of an appropriate control for the immunofluorescence studies and have elected not to improve the English in the discussion.